# Biologic Therapy and Severe Asthma in Children

**DOI:** 10.3390/biomedicines9070760

**Published:** 2021-06-30

**Authors:** Daniele Russo, Paola Di Filippo, Marina Attanasi, Mauro Lizzi, Sabrina Di Pillo, Francesco Chiarelli

**Affiliations:** 1Department of Pediatrics, University of Chieti, 66100 Chieti, Italy; danielerusso1607@gmail.com (D.R.); marina_attanasi@hotmail.it (M.A.); mauro.lizzi.med@gmail.com (M.L.); sabrinadipillo@gmail.com (S.D.P.); chiarelli@unich.it (F.C.); 2Pediatric Allergy and Respiratory Unit, Department of Pediatrics, University of Chieti, 66100 Chieti, Italy

**Keywords:** severe asthma, children, biologic therapy, endotype, phenotype

## Abstract

Severe asthma is a heterogeneous, complex and chronic disease widespread in the pediatric population. According to the recent findings about the different endotypes of asthma in children, each one characterized by specific intracellular molecular pathways, several innovative biologic therapies have been developed. Due to their precise ability to target specific inflammatory type 2 mediators, biologics have revolutionized the care of chronic allergic diseases in the pediatric and adult population. In this review, we aim to provide the latest evidence about the use, indications, efficacy and safety of biologic therapies to treat severe asthma in children and adolescents.

## 1. Introduction

Monoclonal antibodies (mAbs) are monovalent antibodies binding to the same epitope and generating from a single B-lymphocyte clone [1].

Each antibody is composed of two identical heavy chains and two identical light chains assembled to constitute three functional domains: a crystallizable fragment (Fc) and two antigen-binding fragments (Fabs). Fc defines antibody isotype (IgG, IgM, IgA, IgD and IgE) and binds to immune receptors to elicit effector functions. The N-terminal half of the Fab arms contains the variable sequences, which determine the antibody–antigen affinity and thus, its specificity for a molecular target [2].

The exquisite targeted selectivity and, therefore, the lower toxicity due to binding to other targets, led mAbs to be the fastest growing class of drugs on the market [3].

The production of mAbs in mammalian cells is a multistep process. Mammalian cells are the main hosts for mAbs because they correctly perform post-translational modifications; the originally used African green monkey kidney cells were replaced by Chinese hamster ovary and myeloma cells, more suitable for large-scale production. Firstly, the selected cells are transfected: transfection is the integration of the DNA of the gene of interest into the host mammalian genome to obtain mAb-producing clones. Successively, transfected mAbs-producing cells are selected in cultures that allow survival and growth only of cell clones expressing the marker gene product.

These clones are transferred to a second culture medium to produce clonal populations. Repetitive rounds of exposure to higher concentrations of inhibitors of selective markers result in the amplification of the gene of interest expressing the antibody to improve mAb productivity [4]. mAbs were first generated in mice in 1975 using a hybridoma technique which uses the fusion of B-lymphocytes with immortal myeloma cells, generating cells capable of producing antibodies with selective resistance [5]. The first licensed monoclonal antibody was Orthoclone OKT3 (muromonab-CD3), a monoclonal mouse Immunoglobulin G2a (IgG2a) antibody, approved in 1986 and used to prevent kidney transplant rejection [6]. However, chimeric antibodies with decreased immunogenic potential (e.g., Abciximab and Rituximab [7,8,9,10]) were developed to avoid the side-effects (e.g., human anti-mouse antibody response) [11].

Afterwards, humanized antibodies were generated by the complementary-determining region (CDR) grafting technique where non–human antibody CDR sequences were transplanted into a human framework sequence in order to maintain the target specificity [12]. The humanization of antibodies generated a new class of biologic drugs which could be used in pathological conditions requiring a long-term treatment, such as asthma, cancer and auto–immune diseases [12,13,14].

Conventionally, the suffix used in the nomenclature of monoclonal antibodies indicates whether they are murine (–omab), chimeric (–ximab), humanized (–zumab) or fully human (–umab) [15]. Therefore, their activity was first against specific immune cells, such as CD4 or CD3 lymphocytes, to avoid rejection after solid organ transplantation. Successively, mAbs against cytokines involved in inflammatory/autoimmune diseases such as rheumatoid arthritis or inflammatory bowel disease were developed. Furthermore, mAbs with inhibitory activity on specific enzymes, cell surface transporters or signaling molecules essential for tumor or virus growth were developed [3].

The use of monoclonal antibodies is currently extending to non-malignant diseases, including asthma and atopic dermatitis. Nowadays, about thirty mAbs are approved in medical practice and many others are currently being tested in clinical trials representing an innovative therapy for several diseases. However, the results obtained from randomized clinical trials on adults are not always directly transferable to children and adolescents. Therefore, some clinical trials are still in the initial phase for the pediatric population. The advances in genetic sequencing and biomedical research, as well as a more comprehensive understanding of the molecular pathophysiology of asthma in children, carried out the identification of new specific targets in the pediatric population. The aim of this review is to provide a clinical guide on the mAbs used in children with severe asthma, focusing on the main characteristics in terms of applications, safety, efficacy, limitations and future directions in clinical practice.

## 2. Severe Asthma

Asthma is one of the most common chronic, non–communicable diseases in children [16]. The global prevalence is around 5–10% [17], with a wide discrepancy between countries, especially in the pediatric population [18].

Several guidelines state that asthma management should be based on asthma control [19]. The level of asthma control is defined by the clinical symptoms and the ability of therapy to reduce or remove symptoms [20]. In most cases, asthma is controlled by low to medium doses of inhaled corticosteroids (ICS); however, 5–10% of patients continue to suffer from asthma symptoms, frequent exacerbations and reduced lung function despite the use of high doses drugs [21,22].

The definition of severe asthma is still evolving in literature and differs among International Societies [23,24,25]. However, international societies agree in assessing asthma severity based on the treatment level required to achieve and maintain adequate control [26]. According to GINA 2021, severe asthma is defined as uncontrolled asthma despite high dose ICS–long-acting beta2 agonist (LABA) and triggers avoidance, or asthma that requires high dose ICS–LABA to remain controlled.

The Severe Asthma Research Program (SARP) III [27] and U-BIOPRED study [22] found that severe asthma is more common in the 12–15 years age group and in the male sex. Children with severe asthma have frequent asthma exacerbations, lung function impairment, poor quality of life (QoL) and are at high risk of medication-related side effects [28,29].

Before establishing a severe asthma condition, and starting a biologic drug, a multidisciplinary evaluation is essential to exclude conditions mimicking this syndrome [24], comorbidities reducing the response to therapy [30] and uncontrolled asthma (poor adherence, exposure to environmental inhalants, incorrect inhalation technique).

Since asthma is a highly heterogeneous syndrome, phenotypes were proposed to distinguish groups of patients with the same clinical presentation, response to triggers and allergic characteristics [31]. However, the recognition of phenotypes has limited value in predicting the new therapies efficacy, because it does not take into account the underlying pathogenic mechanisms. The development of the endotypes, defined by a biological mechanism that links clinical features with a molecular pathway, allowed the identification of specific biomarkers and thus, targeting therapies [32,33].

Two main endotypes of asthma were identified based on airway inflammation: Type 2 (T2)–high and T2–low asthma. T2–high asthma is the most common in children; it is typically defined by allergic sensitization and eosinophilic airway inflammation, driven by immunoglobulin E (IgE), and interleukin (IL)-4, IL-5 and IL-13. Increased peripheral blood eosinophil counts, blood periostin level, fractional exhaled nitric oxide and allergen specific IgE levels may be used as surrogate markers [34]. T2–low asthma is more frequent in adults and is characterized by normal airway eosinophil and neutrophil counts or increased airway neutrophil counts, sustained by IL-8, IL-17, IL-22 and other T cell-related cytokines, plus epithelial cell-derived cytokines [34].

IgE binds to the high-affinity IgE receptor (FcεR1) on mast cells and basophils, which captures the allergen, resulting in cross-linking of the portion of the IgE antibody and activating synthesis and release of mast cell mediators.

Histamine, prostaglandin D2 (PD2), leukotriene C4 (LTC4), IL-4, IL-5, IL-13, tumor necrosis factor α (TNFα) and chemokines lead to bronchoconstriction and airway inflammation [35,36,37]. In particular:-IL-5 is released by mast cells, T2 and innate lymphoid 2 cells (ILC2s); IL-5 binds to the IL-5 receptor (IL-5R) on eosinophils and basophils, inducing eosinophil proliferation, activation, recruitment and release of cytokines that lead to airway hyperresponsiveness and remodeling [38]-IL–4 and IL-13 are released by mast cells, ILC2s and Th2 cells; they bind to the type 2 receptor complex (IL-4Rα/IL-13Rα1) on airway epithelial and smooth muscle cells, eosinophils, and mast cells. IL-4 also binds to the type 1 receptor complex, consisting of IL-4Rα and a γc chain, which leads to upregulation of T2 responses, downregulation of T1 responses and accumulation of IgE. IL-13 directly affects airway contraction and increases airway mucous production. It also stimulates periostin release from airway epithelial cells, contributing to tissue remodeling [38].

According to these findings about the prevalent endotype of asthma in children, several innovative biologics targeting these specific inflammatory type 2 mediators were developed [34]. Figure 1 shows the point of action of the inhibitory activity of the mentioned biologic drugs.

## 3. Biologic Drugs in Severe Asthma

### 3.1. Omalizumab

Omalizumab is a humanized anti-IgE monoclonal antibody and it was the first mAb with a pediatric indication. Omalizumab binds to IgE Fcε3 segment and prevents their binding to the FcεR1 receptor on mast cells and basophils. Therefore, the rapid clearance of this antibody through the reticuloendothelial system and the decreased expression of FcεR1 on the cell surface result in an attenuated allergic response [39,40]. It was approved by the Food and Drug Administration (FDA) for children of at least six years of age with moderate to severe persistent asthma, uncontrolled symptoms with ICS, a positive perennial aeroallergen sensitization (in vivo or in vitro) and increased serum total IgE levels (IgE > 30 and < 1500 IU/mL) [38]. Although its use above these ranges is not recommended, several studies showed benefits in patients with asthma and IgE levels above these values [34,38].

Omalizumab is administered subcutaneously (SC) every 2–4 weeks, added to step 5 since the 2017 GINA guidelines. Dosage for asthma is determined by a normogram based on bodyweight and pretreatment serum total IgE levels and ranges from 75 to 375 mg [40].

Omalizumab efficacy and safety in children with moderate–severe allergic asthma were widely demonstrated in several trials leading to its approval more than 10 years ago.

The first study was a double-blind, randomized, placebo-controlled study, developed by Milgrom et al. [41]. Three hundred and thirty-four children aged 6–12 years with moderate to severe allergic asthma were randomized for a subcutaneously administered placebo (*N* = 109) or Omalizumab (*N* = 225), using a dosing chart designed to assure a minimum dose of 0.016 mg/kg/IgE (IU/mL) per 4 weeks. In the Omalizumab group, more participants decreased their beclomethasone dipropionate (BDP) dose and the reduction was greater compared to the placebo group (median reduction 100% vs. 66.7%); more participants discontinued the BDP compared to the placebo group (55% vs. 39%); fewer participants had asthma exacerbations (18.2% vs. 38.5%).

In addition, several pediatric studies showed that Omalizumab provided a mild improvement in lung function and a better asthma control by reducing exacerbations (by approximately 40% overall and >60% if blood eosinophil counts are ≥300 cells/µL), hospitalizations, emergency department visits, daily ICS, seasonal exacerbations triggered by respiratory viruses [40,42,43,44,45,46,47,48,49]. However, 34% of severe asthmatic patients may experience poor disease control despite Omalizumab treatment and the causes remain still unclear [34].

Therefore, recent studies found that age > 12 years, asthma exacerbation and hospitalization during the last 6 months, pre-bronchodilator forced expiratory volume in the first second (FEV1) < 90% of predicted, and comorbidities (such as obesity, gastroesophageal reflux, chronic rhinosinusitis, nasal polyps and psychological disorders) were clinical predictors of poor response to Omalizumab treatment [34,45,46]. On the other side, a history of multiple allergies, high total IgE levels, fractional exhaled nitric oxide (FeNO) values and blood eosinophilia (eosinophil counts ≥ 300 cells/μL) were found to be predictors of a better response to Omalizumab treatment [34].

To date, there is no strong evidence that the treatment with Omalizumab really modifies the asthma evolution. Indeed, the optimal duration of therapy and the long-lasting effects are still unclear, and more studies are needed to evaluate good safety and tolerability in children. However, in patients without clinical improvement, it is recommended that patients be reexamined after 16 to 24 weeks of therapy to determine whether Omalizumab therapy should be continued [38,39].

To the best of our knowledge, Omalizumab seems to be safe in real life, with only 0.1–0.2% patients experiencing Omalizumab-associated anaphylaxis, and more common mild local side-effects, such as pain and skin reaction at the injection site. An increase in the risk of developing malignancies was not reported [38].

### 3.2. Mepolizumab

Mepolizumab is a murine humanized IgG1mAb targeting circulating IL-5 and preventing the IL-5/IL-5Rα interaction. It was approved by the FDA and the European Medicines Agency (EMA) as an add-on maintenance therapeutic option for the treatment of severe eosinophilic asthma in patients over 6 years of age [50]. The recommended dose and the timing of administration differ according to the age of the patients: 100 mg/4 weeks subcutaneously in adults and children over 12 years, and 40 mg/4 weeks subcutaneously in children aged 6 to 11 years [50,51].

Currently, standardized response criteria are lacking. However, clinical and laboratory parameters are used as predictive tools to evaluate the therapy efficacy. Blood eosinophil count and the improvement of the lung function are considered as possible parameters of response to Mepolizumab in patients with eosinophilic asthma requiring regular ICS [52,53,54].

There is no validated recommendation on mepolizumab discontinuation. The National Institute for Health and Care Excellence guidelines suggest continuing the treatment if the patient shows more than a 50% reduction in asthma exacerbations after 12 months of Mepolizumab [55].

However, there are also several studies that have shown a worsening of asthma with an increase of peripheral eosinophils after 3–6 months from the interruption of Mepolizumab treatment [56].

To date, the FDA has also approved Mepolizumab for the treatment of eosinophilic granulomatosis with polyangiitis. Moreover, IL-5 blockers are under investigation for other diseases (i.e., eosinophilic esophagitis) even if further well-done studies are needed before obtaining FDA approval.

The most common adverse effects include injection-site reactions, a worsening of asthma, respiratory tract infections, back pain, headaches and fatigue [57].

Two interesting studies, DREAM (Dose Ranging, Efficacy, and Safety with Mepolizumab in Severe Asthma) and MENSA (Mepolizumab as Adjunctive Therapy in Patients with Severe Asthma), investigated Mepolizumab efficacy in patients with eosinophilic asthma older than 12 years. The authors showed a significant clinical improvement in terms of the number of asthma exacerbations by approximately 53% and in terms of an increased FEV1 by approximately 0.1 L in phase 3 of the aforementioned studies. The patients also experienced a decrease in emergency department visits and hospitalizations, as well as an improvement of asthma QoL scores [52,58].

In addition, in the SIRIUS study (the Steroid Reduction with Mepolizumab), Bel et al. found a reduction of about half of cases who used oral corticosteroids (OCS) in the Mepolizumab group compared to the control group involving patients over 16 years of age with severe eosinophilic asthma requiring a daily intake of OCS despite the use of high–dose ICS [59]. Recent studies, such as the COSMOS study, a 52-week, open-label extension trial, combining MENSA and SIRIUS data [60], as well as the Long Term Extension Safety Study of Mepolizumab in Asthmatic Subjects (COLUMBA), have recorded additional benefits of Mepolizumab in patients with severe eosinophilic asthma. In the COLUMBA study [61], patients previously enrolled in the DREAM trial and under an asthma controller medication for 12 or more weeks received 100 mg of subcutaneous Mepolizumab every 4 weeks plus standard of care for a long period (over 156 weeks). The exacerbation rate was 0.74 events/y (weeks 0–156), with a 56% reduction from the off-treatment period between DREAM and COLUMBA. For all patients, the mean Asthma Control Questionnaire 5 score was reduced by 0.47 points, and blood eosinophil counts were reduced by 78%, with similar improvements maintained throughout the study. The immunogenicity profile (8% anti-drug antibodies) was consistent with previous studies [61]. These recent studies have showed the long-term safety and efficacy of Mepolizumab in patients with severe eosinophilic asthma.

### 3.3. Reslizumab

Reslizumab is an IgG4 kappa monoclonal antibody binding circulating IL-5. It was approved in 2016 as an add-on therapy in patients aged ≥18 years with eosinophilic severe asthma [62]. The recommended Reslizumab dosage is 3.0 mg/kg/4 weeks administered intravenously [62,63]. The two main multicenter studies phase III trials that led to the approval of Reslizumab in adults were carried out in a double-blind simultaneous 52-week period, enrolling patients aged 12 to 75 years with a poorly controlled asthma and with an eosinophil count greater than 400 cells/mL. In the treated group, a significant reduction in asthma exacerbations and an improvement in lung function and asthma QoL scores were found compared to placebo group [62,63,64,65]. A recent study has shown that all these benefic effects seem to last for a period of 24 months [66]. The FDA Adverse Event Reporting System (FAERS) Search Strategy showed a favorable long-term safety of Reslizumab in the pediatric population, with only one non-serious side effect (eosinophilic esophagitis) and only one serious event (chronic cholecystitis). Nasopharyngitis, upper respiratory tract infections, sinusitis, urinary respiratory tract infections, worsening of asthma and headache are generally adverse events reported in adults [66]. Although several benefits are reported in literature, further studies are needed to define its clinical efficacy and safety in children.

### 3.4. Benralizumab

Benralizumab has been approved by the FDA and European Medicines Agency (EMA) since 2017. It exerts its therapeutic action binding the isoleucine-61 within domain 1 of the human alpha subunit of the cellular receptor for IL-5 (IL-5Rα). Benralizumab is the most recent monoclonal anti IL-5 Rα antibody authorized for human subjects. Medical indication is for patients over 12 years of age with severe eosinophilic asthma as an add-on treatment. The dosage is 30 mg subcutaneously every 4 weeks (for the first 3 doses), then 30 mg subcutaneously every 8 weeks [67].

Three large multicenter phase III studies investigated the efficacy and safety profiles of Benralizumab in more than 2730 patients from 26 different countries in the world including adults and adolescents. In SIROCCO and CALIMA studies, the authors showed a significant decrement of the number of asthma exacerbations and an improvement of the asthmatic symptoms in patients with severe allergic asthma with eosinophilic count greater than 300 cells/mL compared to controls [68,69]. Nair et al. [70] showed in a RCT, ZONDA study, a drastic decrement of the use of systemic corticosteroids, of the number of asthma flare-ups and hospital admissions for asthma after 12 weeks following Benralizumab treatment in patients with severe asthma compared to controls [70]. The BORA extension trial [71], a randomized, double-blind, parallel-group, phase 3 study, investigated all patients who completed treatment in the SIROCCO, CALIMA or ZONDA trials. Patients were aged 12–75 years and had physician-diagnosed asthma requiring treatment with medium-dosage or high-dosage inhaled corticosteroids and long-acting β2-agonists for at least 12 months before enrolment. Adult and adolescent patients who were previously assigned a placebo in SIROCCO or CALIMA were randomly re-assigned in a 1:1 ratio to receive subcutaneous Benralizumab 30 mg either every 4 weeks (Q4W) or every 8 weeks (Q8W). Patients who were newly assigned Benralizumab Q8W were required to have their first three doses 4 weeks apart. Patients assigned Benralizumab Q8W in BORA received placebo injections at the 4-week interim to ensure masking of regimen assignment. The BORA extensional trial assessed the long-term safety and efficacy of the 2 dosing regimens of Benralizumab over 2 years follow up period [71]. The safety and efficacy profiles for Benralizumab Q8W were similar to, if not numerically better than, the Q4W regimen, with the caveat that this study was not designed to assess differences between treatment groups. These findings are consistent with SIROCCO and CALIMA results and support the use of this regimen for the treatment of patients with severe eosinophilic asthma.

### 3.5. Dupilumab

Dupilumab is a fully humanized monoclonal antibody approved in 2017 in the USA and Europe as an add-on maintenance therapy in adults and adolescents with moderate–severe asthma and/or with moderate–severe atopic dermatitis. Dupilumab binds to the alpha subunit of the IL-4 receptor (IL-4Ra), blocking the signaling mediated by IL-4 and IL-13. These two cytokines are produced by CD4+ Th2 cells and ILC2 and are defined as ‘sister cytokines’ because they bind the two subtypes of the common IL-4Rα [72,73]. IL-4R type I is characterized by the link between the common IL-4Rα with the γc chain and is expressed on the surface of hematopoietic cells. IL-4R type II rises from the pairing of the common IL-4Rα with the binding receptor for IL-13 (IL–13Rα1), forming a specific heterodimeric complex which is located on the surface of both hematopoietic and non-hematopoietic cells.

The binding of IL-4 or IL-13 to their receptors triggers a chain reaction of trans-phosphorylation and activation of receptor subunit-associated Janus family protein kinases (JAKs), including JAK1, JAK2/Tyk and JAK3 associated with the IL-4Rα, IL-13Rα1 and γc chains, respectively. This cascade therefore induces the recruitment of the transcription factor signal transducer and activator of transcription 6 (STAT6). In addition, STAT3 activation via IL-13Rα1 and IRS2 regulation by Socs1/Ubiquitin is another pathway implicated in receptor signaling [74,75].

QUEST and VENTURE are the two phase 3 RCT which led to the approval of Dupilumab in asthma patients with poor symptom control [76,77]. Busse et al. [76] in the QUEST study, involving 1902 patients (6% aged between 12 and 18 years), found a significant reduction in the number of annual severe asthma exacerbations after 52 weeks of therapy in 48% of patients treated with Dupilumab 200 mg and in 46% of those treated with Dupilumab 300 mg. An improvement in the FEV1 parameter was observed in the two treated groups and the result significantly correlated to the peripheral blood eosinophils level at the onset of therapy. Rabe et al. [77] in the VENTURE study included 210 patients over 12 years of age with severe asthma and in treatment with systemic corticosteroids, without considering the baseline value of peripheral blood eosinophils. The aforementioned study’s patients were randomized to receive Dupilumab with a starting dose of 600 mg followed by a dose of 300 mg or placebo every week, for a total period of 24 weeks. In the Dupilumab group, there was a decrease of more than 70% of the systemic steroid dose required, compared to 41.9% of the placebo group; in addition, the patients treated with Dupilumab showed a reduction of 59% in the number of asthma exacerbations compared to the overall population despite the significant decrement of the corticosteroid doses. Furthermore, patients with an eosinophilic count ≥300 cells/mL before starting the treatment showed a decrease of about 71% in the number of asthma exacerbations and a significant improvement of FEV1.

Regarding the safety profile of Dupilumab, the most commonly reported adverse events include injection site reactions, upper respiratory tract infections and headache. After beginning the treatment, a transiently elevated eosinophil count was observed in 4.1–14% of patients [77]. Dupilumab, inhibiting IL-4 and IL-13 signaling and thus the production of chemotactic factors for eosinophils (i.e., eotaxin), blocks the migration of eosinophils to the peripheral tissues without affecting their medullary production, with a consequent transient increase in the count of circulating eosinophils [76,77]. Furthermore, Dupilumab has also shown a significant clinical efficacy in patients with comorbidities, including chronic rhinosinusitis with nasal polyposis, atopic dermatitis and eosinophilic esophagitis, suggesting a common pathway between asthma and those other diseases characterized by high eosinophil levels [78,79].

### 3.6. Tezepelumab

Tezepelumab is one of the latest humanized mAbs under investigation currently. It binds thymic stromal lymphopoietin (TSLP), an epithelial-cell-derived cytokine implicated in the pathogenesis of asthma, preventing the interaction of TSLP with its receptor which is expressed on different immune cells of the type 2 inflammatory cascade [80]. Corren et al. [80] carried out a phase 2 RCT (NCT02054130) to compare Tezepelumab administered at 70 mg every 4 weeks, 210 mg every 4 weeks, 280 mg every 2 weeks or placebo. All patients in treatment with any dose of Tezepelumab showed a significant reduction in the annual rate of asthma exacerbations and an improvement in the pre-bronchodilator FEV1 compared to controls. The finding of the reduction of Th2 biomarkers (eosinophils, FeNO, and IgE) suggests an influence of Tezepelumab on IL-4, IL-5 and IL-13 pathway [80].

To date, in the literature, several studies evaluating the safety, tolerability and efficacy profile of Tezepelumab in adults with severe asthma are ongoing (NAVIGATOR, CASCADE, NCT03688074; DIRECTION, NCT03927157; NCT03347279).

Recently, the NAVIGATOR phase 3 multicenter study enrolled patients aged between 12 and 80 years old, who were randomly assigned to receive Tezepelumab (210 mg) or a placebo subcutaneously every 4 weeks for 52 weeks; the annual rate of asthma exacerbations (primary endpoint) and FEV1 and asthma QoL scores (second endpoints) were assessed. The patients were divided into four subgroups according to the level of blood eosinophils and FeNO values. These two inflammatory biomarkers were used by clinicians to evaluate treatment options (blood levels of eosinophil count (≥300 or <300/uL) and FeNO (≥ 25 or < 25 ppb)) [81].

The NAVIGATOR trial showed a reduction in the annual rate of asthma exacerbations over 52 weeks in patients treated with Tezepelumab compared to controls when added to the standard of care, regardless of blood eosinophil counts, allergy status and FeNO level. Nevertheless, the major benefits were observed in asthmatic patients with high eosinophilic count and FeNO levels; indeed, in patients with baseline blood eosinophil counts ≥ 300 cells/uL and FeNO levels ≥ 25 ppb, Tezepelumab induced a reduction of 77% in the annual rate of asthma exacerbations and a reduction of 85% of asthma exacerbations requiring hospitalization compared to placebo. Further benefits of Tezepelumab were demonstrated in terms of improvement of lung function parameters, asthma control and health related QoL [81].

Nowadays, other phase 3 RCTs are ongoing to evaluate the corticosteroids sparing effect of Tezepelumab in adults with severe asthma (SOURCE, NCT03406078).

In recent RCTs, the drug-related serious adverse events were pneumonia and stroke (in the low-dose Tezepelumab group) and Guillain–Barré syndrome (in the medium-dose Tezepelumab group). Neither investigational product-related anaphylactic reactions nor the identification of neutralizing antibodies were reported [80]. 

Table 1 illustrates biologic drugs used in children with severe asthma.

## 4. Conclusions

Recent insights in the pathophysiology of allergic disorders have allowed the identifying of novel therapeutic strategies for the treatment of severe asthma in the pediatric population, aiming to positively change the natural history of allergies and improve the QoL of children. Indeed, the characterization of phenotype and, recently, of endotype have allowed the development of several biologic drugs targeting specific intracellular pathways of the inflammatory allergic cascade. However, the identification of the ideal drug, as well as the optimization of the dosage and duration of the treatment, are still a matter of debate mostly due to the wide variability in response to the treatment.

To better understand which biologic drug can fit with the endotype of asthma investigated, potential factors, such as inflammatory biomarkers, convenience for the patient, comorbidities and pharmacoeconomic aspect, should be considered. Indeed, the most recent studies suggest as therapy of choice Omalizumab use for T2 high allergic asthma and IL-5 blockers or Dupilumab use for T2 high non-allergic eosinophilic asthma considering the method of administration, the frequency of treatment and the clinical improvement as well. According to the literature, we suggest a flowchart to help physicians to choose the best biologic therapy in severe asthma (Figure 2).

Adherence to the treatment is one of the key points in choosing the best therapy for every asthmatic child. All aforementioned biologic drugs, except for Reslizumab, are administered SC. Dupilumab has got the further benefit of home administration despite requiring more frequent administrations (every 2 weeks). Benralizumab has the positive aspect to be administered every 8 weeks after the first 3 doses avoiding wasting time from school and other duties.

However, to date, further RCTs are needed to compare the different biologic drugs regarding the efficacy and safety, to address the physicians to choose the best tailored treatment for severe asthma mostly in children in clinical practice.

## Figures and Tables

**Figure 1 biomedicines-09-00760-f001:**
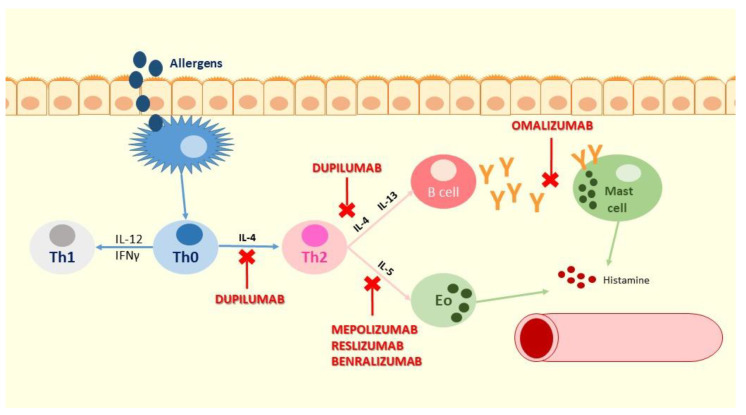
The point of action of the inhibitory activity of the mentioned biologic drugs.

**Figure 2 biomedicines-09-00760-f002:**
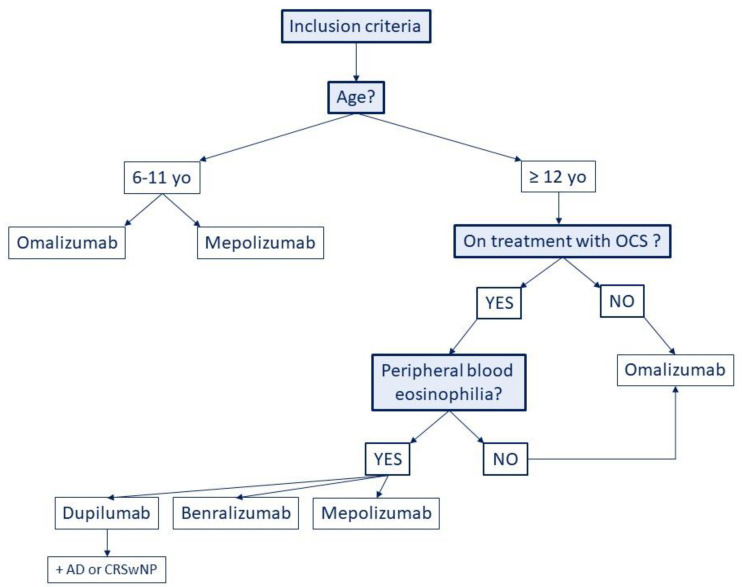
A flowchart to help physicians to choose the best biologic therapy in severe asthma. Yo = years old; OCS = oral corticosteroids; DA = atopic dermatitis; CRSwNP = chronic rhinosinusitis with nasal polyposis.

**Table 1 biomedicines-09-00760-t001:** Main Biologic Drugs currently approved in the pediatric population.

Biological Drug	Structure	Action	Dosage	Age (Years)	References
**Omalizumab**	Humanized IgG1	Anti-IgE	In **moderate to severe allergic asthma:** SC 75 to 375 mg SC/2-4 wk In **CRSwNP** and **CSU**	≥6 (AIFA) ≥6 (EMA) ≥6 (FDA) ≥18 in CRSwNP	[39,40]
**Mepolizumab**	Humanized IgG1	Anti-IL-5	In **severe eosinophilic asthma:** SC 100 mg/4 wk ≥ 12 yo SC 40 mg/4 wk ≥ 6 yo	≥6 (AIFA) ≥6 (EMA) ≥6 (FDA)	[50,51]
**Reslizumab**	Humanized IgG4	Anti-IL-5	In **severe eosinophilic asthma:** IV 3.0 mg/kg/4 wk	≥18 (AIFA) ≥18 (EMA) ≥18 (FDA)	[62,63]
**Benralizumab**	Humanized IgG1	Anti-IL-5Rα	In **severe eosinophilic asthma:** SC 30 mg/4 wk	≥18 (AIFA) ≥18 (EMA) ≥12 (FDA)	[67]
**Dupilumab**	Human IgG4	Anti-IL-4Rα	In **moderate-to-severe eosinophilic asthma, CRSwNP & moderate-to-severe atopic dermatitis** Initial dose of SC 600 mg, then 300 mg/2 wk in OCS treat. or + AD initial dose of SC 400 mg, then 200 mg/2 wk	≥12 (AIFA) ≥12 (EMA) ≥12 in asthma ≥6 DA (FDA)	[72,73]
**Tezepelumab**	Human IgG2	Anti-TSLP	In **severe asthma**, especially with: - high blood eosinophil counts (≥300 cells per microliter) and - FeNO levels (≥ 25 parts per billion) SC 210 mg/4 wk	Phase 3 RCTs (NAVIGATOR, SOURCE) ongoing ≥ 12	[80,81]

SC = subcutaneously; wk = week; yo = years old; IV = intravenous; CRSwNP = chronic rhinosinusitis with nasal polyposis; CSU = chronic spontaneous urticaria; AIFA = Agenzia Italiana del Farmaco; EMA = European Medicines Agency; FDA = Food and Drug Administration; IV = intravenously; OCS = oral corticosteroids; AD = atopic dermatitis; FeNO = fractional exhaled nitric oxide.

## Data Availability

Not applicable.

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
