# Peer review of "Biologic Therapy and Severe Asthma in Children"

_biomedicines, 2021, doi:10.3390/biomedicines9070760_

Round 1

Reviewer 1 Report

Biomedicines

Manuscript ID: 1281065

Type of manuscript: Review

Title: Biologic Therapy in Pediatric Severe Asthma

The Authors present an up-to-date review of the data on the currently available biologic therapies in pediatric severe asthma, including their safety and efficacy. The paper is interesting for physicians interested in this field as it is supported by updated list of related studies. It was a very good idea to include the flowchart - Fig. 2 showing recommendations how to select the best treatment for severe asthma in children in clinical practice.

Generally, this is very good and exhaustive review of the current knowledge in this field, nicely written and in good English, which I recommend for publication in the present from.

Author Response

Thank you very much for your comment.

Reviewer 2 Report

In this review, Russo and collaborators perform an extensively and detailed overview of the different biological treatment for childhood severe asthma. The article is well structured and provided sufficient and detailed information about this topic. They describe in depth anti-IL5 therapies and others such as omalizumab (anti-IgE), tezepelumab (anti TSLP) and dupilumab (anti-IL-4/IL-13). I only have several minor comments:

  1. Table 1: Please, provide the head for each column (i.e.: Biological drug, Target molecule, etc.). Also you must add a new column with the references for each treatment.
  2. Figure 1 must be place before in text (just before section 3, Line 142).
  3. Lines 268-269: “Benralizumab is the most recent monoclonal anti IL–5 antibody authorized for the human subjects” is not quite correct. Benralizumab is not anti-IL-5 antibody, is anti-IL-5Rα antibody. Please, reformulate this sentence.

Author Response

We thank the reviewer for his/her relevant comments.

  1. We provided the head of the table and we added a new column with the references.
  2. We placed figure 3 before section 3, as suggested
  3. We reformulated the sentence, as suggested